# Longitudinal Effects of Screen Time on Depressive Symptoms among Swedish Adolescents: The Moderating and Mediating Role of Coping Engagement Behavior

**DOI:** 10.3390/ijerph20043771

**Published:** 2023-02-20

**Authors:** Sebastian Hökby, Joakim Westerlund, Jesper Alvarsson, Vladimir Carli, Gergö Hadlaczky

**Affiliations:** 1National Centre for Suicide Research and Prevention (NASP), Department of Learning, Informatics, Management and Ethics (LIME), Karolinska Institutet, 171 77 Stockholm, Sweden; 2National Centre for Suicide Research and Prevention (NASP), Centre for Health Economics, Informatics and Health Services Research (CHIS), Stockholm Health Care Services, 171 77 Stockholm, Sweden; 3Department of Psychology, Stockholm University, 106 91 Stockholm, Sweden; 4Stockholm Centre for Health and Social Change (SCOHOST), Department of Psychology, School of Social Sciences, Södertörn University, 141 89 Huddinge, Sweden

**Keywords:** screen time, internet use, depression, coping behaviors, emotional regulation, problem solving, social skills, adolescent development, longitudinal studies, public health

## Abstract

Studies suggest that hourly digital screen time increases adolescents’ depressive symptoms and emotional regulation difficulties. However, causal mechanisms behind such associations remain unclear. We hypothesized that problem-focused and/or emotion-focused engagement coping moderates and possibly mediates this association over time. Questionnaire data were collected in three waves from a representative sample of Swedish adolescents (0, 3 and 12 months; *n* = 4793; 51% boys; 99% aged 13–15). Generalized Estimating Equations estimated the main effects and moderation effects, and structural regression estimated the mediation pathways. The results showed that problem-focused coping had a main effect on future depression (*b* = 0.030; *p* < 0.001) and moderated the effect of screen time (*b* = 0.009; *p* < 0.01). The effect size of this moderation was maximum 3.4 BDI-II scores. The mediation results corroborated the finding that future depression was only indirectly correlated with baseline screen time, conditional upon intermittent problem-coping interference (C’-path: *Std. beta* = 0.001; *p* = 0.018). The data did not support direct effects, emotion-focused coping effects, or reversed causality. We conclude that hourly screen time can increase depressive symptoms in adolescent populations through interferences with problem-focused coping and other emotional regulation behaviors. Preventive programs could target coping interferences to improve public health. We discuss psychological models of why screen time may interfere with coping, including displacement effects and echo chamber phenomena.

## 1. Introduction

Mental disorders and mental health problems among children, adolescents and young adults are important public health concerns. These detrimental states are the greatest cause of years lived with disability (YLDs) among European youth aged 10–24 years [1]. According to the Global Burden of Disease Study 2019, the greatest burden of mental disorders in this age group comes from depression and anxiety; the estimated prevalences of anxiety and depression disorders were 6.5% and 3.1% respectively [1]. It is typical for these disorders to have their onset during adolescence [2,3]. In Sweden, the prevalences resemble the European averages but are increasing annually in the adolescent population, leading to escalating health care consumption costs related to these disorders [1,4,5]. In the long term, depression and anxiety also contribute to young people’s risk of poor academic achievement, labor market integration, economic stability, as well as suicide and other causes of early death [6]. Societal changes that increase the risk of distress should be considered as possible explanations for such worrying trends, for example, stressors related to screen time, social media use and other screen contents.

“Problematic Use of the Internet” (PUI) is an umbrella term encompassing a range of potentially detrimental internet-related behaviors, such as excessive or compulsive gaming and social media use [7,8]. Risk factors for PUI are typically also risk factors for suicide [9,10,11], including low socioeconomic status [12,13], social isolation and loneliness [14], low social functioning and low self-esteem [14,15], familial factors such as parental supervision [16,17], and poor school relationships and academic achievements [18]. Screen time is intimately related to PUI yet conceptually different, as screen time is not necessarily maladaptive [19,20]. However, studies suggest that even moderate amounts of screen time can have negative health consequences. For instance, screen time may displace more healthy behaviors [21,22], or reinforce unhealthy behaviors since some screen time activities tend to provide the user with information that confirms rather than challenges existing beliefs [23,24,25]. The present study focused on screen time, and most studies identified by recent systematic reviews support a general association between screen time and mental health problems, particularly depression but also anxiety, stress, various addiction problems, self-harm and sleep quality; however the causal direction of these effects are not well-established [7,21,26]. For instance, sleep problems (particularly sleep quality and duration) have been hypothesized to act as mediators or moderators in the relationship between screen time and mental health problems [27,28,29,30,31,32]. In a previous study [27], we found that depression, anxiety and stress symptoms were more likely to occur in subjects reporting sleep-related consequences of screen use, even when screen time, screen content and other control variables were held constant. Evidence from meta-analyses corroborate the importance of sleep in the context of screen time, although some studies have suggested that younger children are more susceptible to screen-related sleep problems [30], while other studies have suggested that screen time impacts mental health more severely in adolescents than younger children [11]. Age-related effects might be related to neurophysiological development during certain life phases, and higher sensitivity to sleep problems might confer health risks proportional to the extent that sleep influences mental health [30]. Therefore, the possibility to generalize findings across different age groups might be limited in this research field.

Early studies that investigated associations between screen time (or PUI) and mental health problems were mainly cross-sectional. Such designs only generate limited evidence because there could be reversed causality that accounts for such associations; thus, longitudinal studies have become more pertinent in this field [21]. Longitudinal studies are better equipped to examine temporal and causal mechanisms, and the present study draws upon this notion. The current research indicates that associations between screen time and depressive symptoms generally have small effect sizes. A recent systematic review suggested that most longitudinal studies of reasonably high quality reported an effect size equivalent to approximately *r* = 0.10 [21]. The same review also suggested that the association is stronger for depression compared to anxiety and other internalizing symptoms, and that screen time predicts future depression more consistently than depression predicts future screen time.

Furthermore, the health effects of screen time seem to differ between genders, where girls and women seem to be more susceptible to negative health consequences compared to boys and men [11,33]. Previous studies have further noted that moderating and mediating effects could be larger among girls compared to boys, although this depends on which health measurement is used [11,31,34,35]. Explanations for such gender differences are not entirely clear, but it is safe to say that screen-related behaviors and preferences generally differ between the genders, especially with regards to social media use. A systematic review [35] of 43 studies found that, out of 20 studies wherein significant gender differences were observed, 14 studies found female gender to be a significant effect moderator of that association. A meta-analysis [34] further showed that pooled associations between screen-time-based sedentary behaviors and depressive symptoms were elevated among girls but were non-significant and heterogenous among boys, while simultaneously showing that social media use predicted well-being differently depending on gender (58% mediation among girls and 12% among boys).

Escapism, lack of assertiveness, emotion regulation difficulties and various forms of avoidant coping are associated with PUI, problematic smartphone use and gaming, as well as other behavioral problems [15,36,37,38]. However, the correlates of screen time might differ between adolescents and other age groups. For example, a study that found avoidance coping to mediate the mental health effects of PUI, simultaneously found that avoidance coping was more characteristic of PUI in adults compared to adolescents [39]. In comparison to adults, adolescents with PUI used discussion forums more frequently and had higher rumination levels and lower levels of self-care. Moreover, a study on gamers [40] found that only four out of eight investigated coping strategies moderated the association between psychiatric symptoms and internet gaming disorder severity, and there were no substantial differences between recreational versus professional players. Significantly moderating coping strategies included self-blame/self-distraction and denial, which moderated psychiatric symptoms when addiction was relatively high. Emotional/social support and active coping were also significant but moderated psychiatric symptoms when addiction was relatively low [40]. A similar study on gamers [19] found that withdrawal/resignation coping styles were statistically more important mediators of depressive states in addicted gamers, while approach coping and offline social support mediated depressive states even in non-addicted gamers. Moreover, the study found that offline social support was a more robust mediator than online social support in explaining the association between internet gaming disorder and depression, anxiety and stress [19]. However, these cross-sectional studies were focused on PUI, and mainly gaming content, and do not necessarily generalize to how mental health is impacted by general screen time in the normal adolescent population. Neither can they be generalized to convey to what extent such health effects are moderated or mediated by different kinds of engaging, approaching or proactive coping behaviors (in contrast to, or in addition to, “disengaging” or avoidant coping styles). Rather, the studies indicate that it is possible that different forms of engagement coping may play different roles in the normal population compared to populations characterized by PUI or problematic gaming.

Whether avoidant or approaching in nature, dichotomizing problem-focused versus emotion-focused coping is a common approach [41,42,43]. However, regardless of how various coping strategies are grouped or categorized, age and gender differences are typically observed in the normal population. Age and gender differences in problem-focused and emotion-focused coping are likely to have innate (dispositional, genetic, biological), situational and sociocultural explanations [41,42,43]. In absolute terms, women tend to engage more in most coping strategies compared to men, including emotion-focused and problem-focused coping. According to a meta-analysis, women’s tendency to express emotions and to seek social support is especially robust and homogenous across studies; and it is only in the relative sense that men seem more prone to engage in problem-focused coping rather than emotion-focused coping [41]. Women’s tendency to use emotional coping has been suggested to explain why they also report more symptoms of depression and anxiety (seemingly implying that problem coping is a more constructive way of managing stress) [44]. On the other hand, female gender is also associated with certain personality traits and coping cognitions, such as higher levels of neuroticism and stress appraisal. Thus, an alternative explanation to the gender difference in psychopathology is that women perceive many situations as more stressful than men, conferring a higher need for emotion-focused coping (which might be more constructive if a stressor or problem cannot be controlled or solved) [42,44]. Moreover, the link between personality traits and coping styles is stronger among young people, and typically diminishes with age [42,43]. In the current study of adolescent boys and girls, we examined problem-focused and emotion-focused coping styles separately, partly due to expected gender differences, and partly to investigate how screen time may affect depression levels depending on individual characteristics in coping styles.

This study aimed to examine the longitudinal association between screen time and depressive symptoms among Swedish adolescents, with particular attention as to how the association is influenced by two different coping styles: problem-focused engagement coping and emotion-focused engagement coping. The formal hypothesis was that screen time levels would predict depression levels over time, with a negative moderation relation to one or both coping styles. A secondary aim was to examine whether coping was a mediator of this association, and whether such an association was bidirectional.

## 2. Methods and Materials

### 2.1. Study Design, Participants and Ethical Considerations

This study used longitudinal self-report data from health questionnaires, originally collected from a large cohort of Swedish adolescents (baseline N = 10,299; 48.8% girls; T1 = baseline, T2 = three-month follow-up; T3 = twelve-month follow-up). The data were collected within a school-based cluster-randomized controlled trial aiming to evaluate health effects of the psychoeducational and mental-health-promoting program “Youth Aware of Mental health“ (YAM; [45]). Participants were recruited from *n* = 116 elementary schools in Stockholm County, located in 24 of the 26 Stockholm municipalities (excluding Nykvarn and Vaxholm municipality; the principals or boards of eligible schools in these areas declined voluntary participation). Approximately 51% of participants were aged 14 years at baseline, and 99% were aged between 13 and 15 years (*n* = 151 reported younger/older age or had missing age data).

Electronic questionnaires were administered using computer tablets in a classroom setting. For ethical reasons, the questionnaires were de-identified but not completely anonymous, and the study design incorporated an emergency intervention for individuals at high risk of suicidal behavior (based on the Paykel Suicide Scale; [46,47]). Participants’ identities were encrypted in the questionnaire using unique participation codes, which enabled questionnaire data to be connected longitudinally, and making contact information available to emergency intervention staff. The participants provided personal information on a separate sheet of paper, which, in turn, provided them with instructions about the data collection, as well as their pseudonymized questionnaire log-in details. Written consent was obtained from all participants, and from their legal guardians in cases where the participant was under 15 years of age. Participants were free to ask questions about the survey and questionnaire items to a supervising data collector. The study was approved by the regional ethics committee in Stockholm (ref.no/ 2016/2175-31/5).

The present analyses only included subjects from the control schools of the cluster-randomized controlled trial. Subjects that received either YAM intervention (*n* = 5192) or emergency intervention within the control group (*n* = 236) were excluded because their mental health trajectories could have been affected by the intervention(s). For instance, such effects could appear because the YAM program promotes coping behaviors and teaches recognition of depressive symptoms. After removing these cases and cases with missing data regarding gender, socioeconomic status or any of the four key variables (screen time, depression, problem-focused and emotion-focused engagement coping), the final longitudinal sample size became N = 4793 (49.3% females).

### 2.2. Procedures and Measurements

At each wave of data collection, four key variables were calculated from multiple items in the questionnaire data: screen time scores (called “screen time” hereon after), depressive symptoms (BDI-II scores), problem-focused engagement coping scores (called “PFE” hereon after) and emotion-focused engagement coping scores (called ”EFE” hereon after). Additional information regarding gender and perceived socioeconomic status was collected at baseline and used as control variables. Socioeconomic status was measured subjectively using a single item (“If you think about the past week, did you have enough money to do the same things as your friends?”). The item was measured on a 1–5 Likert scale (1 = Never; 5 = Always). This item is comparable to previously utilized estimates of socioeconomic status among Swedish youth [48] (p. 20).

Depressive symptoms were measured using the Swedish translation of Beck’s Depression Inventory—Second edition (BDI-II; [49,50]). The original scale contains 21 items, each measured on a 0–3 Likert scale, and the range (min-max) of the total score is thus 0–63 (example item: “I am so sad or unhappy that I can’t stand it”). Beck’s suggested cut-off for “Mild” depression is ≥14 scores [49]. For purposes of statistical modeling, we added a constant of 1 to each item, whereafter scores were calculated as the mean of all 21 items. Our scale thus had a range (min-max) of 1–4. Internal consistency, in terms of Cronbach’s alpha and McDonald’s hierarchical omega and total omega, was acceptable at all timepoints (α_Cronbach_ > 0.90; ω_Hierarchical_ > 0.75; ω_Total_ > 0.93).

Screen time scores were calculated as the mean of three individual items, designed by the authors, that aimed to measure the frequency of daily hours related to screen-based leisure activities (example item: “During a normal day, how much time do you spend watching TV, playing computer games/console games or surf the internet?”). The first item was measured on a 1–6 ordinal scale (response options were: <1 h, 1–2 h, 3–4 h, 5–6 h, 7–8 h or >8 h). The other two items measured internet time during weekdays and weekends, respectively (“How many hours per day, on average, do you use the internet for leisure activities?”), and whereas the original responses were provided on a 24 h integer scale, they were later recoded into the same 1–6 ordinal Likert scale as the first item. Acceptable internal consistency was achieved at all timepoints (α_Cronbach_ > 0.77; ω_Hierarchical_ > 0.79; ω_Total_ = 1.0).

Coping was measured with the 32-item adaptation of the Coping Strategies Inventory (CSI) [51], utilizing two second-order subscales related to engagement coping: “Problem-focused engagement coping” (PFE = problem solving + cognitive restructuring; α_Cronbach_ > 0.88; ω_Hierarchical_ > 0.69; ω_Total_ > 0.94) and “Emotion-Focused engagement coping” (EFE = emotional expression + social contact; α_Cronbach_ > 0.88; ω_Hierarchical_ > 0.69; ω_Total_ > 0.94). The PFE and EFE factor scores were calculated as the mean of their eight respective items, and the scales were reversed so that higher scores reflected less coping engagement (i.e., a “risk factor”). Example items of PFE are “I made a plan of action and followed it” and “I looked at things in a different light and tried to make the best of what was available”. Example items of EFE are “I let my feelings out somehow” and “I talked to someone about how I was feeling”. The coping scale items were answered in relation to a self-reported, recently occurring stressful situation, and school-related stress was by far the most frequently mentioned stressor. The CSI scale originally consists of 72 items [52] and has a psychometrically established hierarchical factor structure [53]. It is designed to measure both engagement and disengagement coping, grouped into 8 first-order subscales, 4 second-order subscales and 2 third-order subscales; each item measured on a 1–5 Likert scale. The CSI-72 has been adapted in several ways to fit different research needs, for example, by reducing the number of items while maintaining the same factor structure. For example, the CSI-32 item version [51] measures four coping engagement factors by using only four items per subscale (Problem Solving: items 1, 9, 17, 25; Cognitive Restructuring: items 2, 10, 18, 26; Express Emotions: items 3, 11, 19, 27; Social Contact: items 4, 12, 20, 28). These items are taken directly from the CSI-72 version (Problem Solving: items 25, 33, 41, 57; Cognitive Restructuring: items 10, 26, 34, 42; Express Emotions: items 27, 35, 43, 59; Social Contact: items 12, 20, 36, 52), while excluding five items from each original subscale [52]. The CSI has been validated using even fewer items, for example, a 16-item version [54] including its Swedish translation [55], and a 15-item version [56].

### 2.3. Statistical Analyses

Analyses were conducted in IBM SPSS 28, if not stated otherwise. Scale reliability tests and structural regression analysis were conducted in *R*, through *R*-studio. All tests used a conventional significance level (two-tailed α = 0.05). The original dataset contained 31% missing values due to occasional non-responses to questionnaire items or due to non-participation at follow-up waves. Little’s ꭓ^2^ test of MCAR was significant (*p* < 0.001), suggesting that data were not missing completely at random. To maximize statistical power, ten multiple imputation datasets were generated to replace missing data, wherein missing values were imputed through fully conditional specification using the Markov Chain Monte Carlo (MCMC) method, and where Predictive Mean Matching (PMM) was used to maximize the plausibility of the imputed values [57].

Analyses of internal consistency (values reported in Section 2.2) were conducted in *R* (v. 4.1.2) and *R*-studio (v. 787277e, 2022-07-22) using the Psych package (v. 2.2.5) with the omega function (3 factors, principal components, 1 iteration). Detailed reliability statistics are reported in Appendix A (Table A1), and values indicate that the internal consistency was acceptable for all scales in terms of alpha and total omega. According to the cut-off values suggested by Kalkbrenner (2021) [58], acceptable internal consistency is indicated by alpha values of ≥0.70, and total omega values of ≥0.65. Optimal hierarchical omega values (≥0.80) were not achieved in all instances, particularly not for the coping scales (the lowest values of 0.69 were observed for PFE and EFE at baseline). However, this is not unexpected due to the hierarchical structure of the CSI scale.

The formal hypothesis was tested with Generalized Estimating Equations (GEE) [59,60]. The GEE model utilized an unstructured correlation matrix that was estimated independently for each of the multiple imputation datasets (on average: T1 × T2 = 0.61; T1 × T3 = 0.51; T2 × T3 = 0.64). The dependent variable was depression scores, modelled with a gamma (γ) distribution through a log link to minimize the effects of the positively skewed distribution (Skewness = 1.7; Kurtosis = 3.6). The model estimated the main effects of time, gender, socioeconomic status, screen time, PFE coping and EFE coping. The model also estimated two coping-related interactions (i.e., time × screen time × PFE, and time × screen time × EFE, respectively) to address the main hypothesis (“screen time levels would predict depression levels over time, with a negative moderation relation to one or both coping styles”). The final result was obtained by pooling the regression coefficients from all datasets. We planned to carry out structural regression (pathway) analysis to examine the interaction terms post hoc if statistical significance was indicated by the GEE model.

Because GEE analysis uses (quasi) maximum likelihood estimation, as opposed to sum of squares, the overall model cannot be converted into a proportion of explained variance [51] (p. 38). Only information criteria can be used to determine relative model fit. Moreover, since the GEE model used a gamma distribution with log link, it did not produce regression coefficients that are intuitive to interpret in terms of effect sizes. The unstandardized beta coefficients have no clear interpretation as they measure depression scores on a logarithmic scale, and the exponentiated beta coefficients, Exp(B), are not odds ratios. Instead, the predicted depression score for a given set of coefficient values (i.e., the Predicted Value of Mean of Response; PVMR) can be obtained by calculating the exponent of the entire regression formula. Subsequently, the effect of a single variable, in terms of raw depression scores, can be demonstrated by changing its coefficient value and calculating the entire formula again, and subtracting the new PVMR from the previous one [60] (p. 156). In the present study, the GEE regression formula was used to produce a PVMR value for each individual subject at each timepoint. Subsequently, the pooled mean PVMR for subjects with a certain value (e.g., PFE = 2) was subtracted from the pooled mean PVMR for subjects reporting a reference value (e.g., PFE = 1). For categorical variables where the increments are discrete, the exact PVMR difference is reported. For continuous variables, the differences are not equal across all 1.0-unit increments, so instead the average increment difference is reported, along with the most extreme possible (Max-Min) PVMR difference. For example, in case of SES, PFE coping and EFE coping (scales that range 1–5), an increment from 1 to 2 moves the predicted raw depression score by a value that is different compared to an increment from 2 to 3; and the Max-Min refers to a difference in PVMR when the scale moves from 5 to 1.

The structural regression (pathway) analysis was conducted using the Lavaan package [61,62] in *R*-studio. The model was saturated with zero degrees of freedom (*df* = 0) and estimated using maximum likelihood, run through the semTools (v. 0.5–6) runMI function (fun = SEM, bootstrap (*n* = 50,000, bca.simple)) to handle the ten multiple imputation datasets. The analysis of regression paths assessed the strength of the mediated relations between screen time, PFE and depressive symptoms across the three waves. As illustrated in Figure 1, a recursive model was used to simultaneously examine the expected directionality as well as the reversed causal direction. The model included covariances among the disturbance terms (measurement errors) at T2 and T3, respectively. The possible presence of non-normal effects was managed by running 50,000 bootstraps.

## 3. Results

Descriptive statistics for screen time, PFE coping, EFE coping and depressive symptoms are shown in Figure 2 below, separately for the two genders and the three waves of data collection. The 95% confidence intervals around these estimates can be used to draw some important inferences. For example, girls reported more depressive symptoms than boys, and both genders had decreasing symptoms over time. The average depression score in the sample was *M* = 1.38 (*SD* = 0.39; pooled *M* = 1.35; pooled standard error, *SEM* = 0.003). This mean corresponds to 7.98 BDI-II scores. Another finding was that boys engaged more in PFE coping than EFE coping, while girls engaged in both coping behaviors to the same extent. Both genders had approximately the same screen time scores (the mean response corresponds to 3–4 h of daily leisure screen time). Screen time scores also increased during the course of the study, especially during the last nine months. Perceived socioeconomic status was only measured at baseline (not shown in Figure 2), and a descriptive analysis showed that most participants reported having enough money to do the same things as their friends/peers (Never = 1%; Seldom = 3%; Sometimes = 8%; Often = 22%; Always = 66%).

Bivariate Pearson correlations between the four key variables are reported in Appendix A (Table A2) and can be summarized as follows: The T1 × T1 (and corresponding T1 × T3) variables screen time and depression correlated *r* = 0.21 (*r* = 0.12); screen time and PFE correlated *r* = 15 (*r* = 0.11); screen time and EFE correlated *r* = 0.08 (*r* = 0.06); PFE and EFE correlated *r* = 0.59 (*r* = 0.25); PFE and depression correlated *r* = 0.39 (*r* = 0.24); and EFE and depression correlated *r* = 0.17 (*r* = 0.11). All correlations were significant at the 0.05 alpha level and explained between 1% and 15% variance in depressive symptoms.

### 3.1. Interaction between Screen Time, Depressive Symptoms and Coping Styles

First, when the Generalized Estimating Equations (GEE) model was run without specifying any interaction terms, all predictors (time, gender, SES, screen time, PFE, EFE) were statistically significant (*p* < 0.001). However, the main model, which included the two interaction terms (time × screen time × PFE and time × screen time × EFE), rendered some variables non-significant (results shown in Table 1). Depressive symptoms were positively associated with female gender (*b* = 0.131; *p* < 0.001) and lower socioeconomic status (*b* = 0.063; *p* < 0.001). Importantly, however, we found no main effect of screen time (*p* = 0.469). Regarding coping styles, the model did not yield any significant effects of emotion-focused coping (EFE; *p* = 0.895), but still found a significant main effect of problem-focused coping (PFE; *b* = 0.03; *p* < 0.001). In a similar fashion, the model yielded a significant time × screen time × PFE interaction (*b* = 0.01; *p* < 0.01), but no significant time × screen time × EFE interaction (*p* > 0.262). The former interaction term implies that screen time was associated with depressive symptoms when it coincided with poorer PFE coping, but not when PFE was held constant (the bivariate longitudinal regression lines and their 95% confidence intervals are illustrated in Appendix A, Figure A1 and Figure A2). Our hypothesis that “screen time levels would predict depression levels over time, with a negative moderation relation to one or both coping styles” was thus supported by the data when it comes to PFE coping but not EFE coping.

Sensitivity analyses showed that the GEE model produced similar estimates regardless of whether the analysis was applied to the original dataset with missing values, or to the multiple imputation datasets. For clarity, only the results for the pooled imputed datasets are presented in Table 1. Moreover, considering the observed gender difference in coping styles (Figure 2), we also ran the GEE model separately for boys and girls, but both analyses gave about the same pattern of results (cutting the sample size in half according to gender increased the *p*-values, but the unstandardized beta coefficients remained approximately the same). Consequently, we only conducted the post hoc examination of the time × screen time × PFE interaction for both genders together.

### 3.2. Effect Size Interpretation

Table 1 displays coefficient effect sizes in terms of PVMR differences, which can be interpreted similarly to unstandardized beta coefficients in linear (OLS) regression. For example, the predicted raw depression score was +0.20 higher for girls (PVMR = 1.45) compared to the reference category of boys (PVMR = 1.25). Furthermore, increasing PFE by 1.0 unit increased predicted depression by approximately +0.097 raw scores, and increasing it by 5.0 units (maximum) increased predicted depression by +0.292 raw depression scores. Regarding the time × screen time × PFE interaction, a 1.0-unit increment can be translated into about +0.065 depression scores (depending on the time point; in this case at T3), and the most extreme possible difference (moving from 1 to 30) can be translated into +0.447 raw depression scores (at T3). Recall that the PFE term is scaled differently (1–5) from the interaction term (1–30), so rather than comparing their 1.0-unit effect sizes, it is more interesting to compare them with respect to the Max-Min effect sizes (+0.292 vs. +0.447). It is also important to note that, even though these PVMR differences measure depression on a scale from 1 to 4, they can be multiplied by 21 (i.e., the number of BDI-II items) to convert effect sizes back to the original BDI-II scale—and these scores can then be compared to Beck’s suggested cut-off score for mild depression (≥14 scores). For example, the Max-Min difference for PFE (+0.292) corresponds to +6.13 BDI-II scores, while the Max-Min difference for the interaction effect (+0.447) corresponds to +9.45 BDI-II scores. This means that the depressive effect of PFE can potentially be extended by +3.36 BDI-II scores if “screen time” is held at its maximum (>8 h per day).

To illustrate the screen time × PFE interaction in even greater detail, a set of 20 score combinations are presented as a heatmap in Table 2, where PVMR values are averaged across all three timepoints. Each cell represents the pooled mean PVMR for cases reporting a particular combination of values, and the heat pattern can be said to be representative of the interaction effect in the entire sample. The Max-Min difference in Table 2 is a PVMR difference of +0.45 depression scores, which again corresponds to +9.45 BDI-II scores (take 1.63 in the lower right cell, subtract 1.18 in the upper left cell, then multiply the result by 21). Note that some of the combinations in Table 2 have few cases (smallest cell: *n* = 98) meaning that their pooled mean PVMR values are calculated from a smaller, less representative, number of cases compared to other cells (largest cell: *n* = 1659).

### 3.3. Pathway Analysis (Post Hoc)

The post hoc test was primarily used to determine whether the observed interaction effect was a mediation effect. Since the GEE analysis showed that the time × screen time × PFE interaction was significant when adjusted for all other effects (gender, socioeconomic status, EFE coping), the post hoc model was unadjusted. Instead, the structural regression model aimed at disentangling the pathways through which screen time at T1 predicted depressive symptoms at T3 (and vice versa). Whereas the analysis output encompassed a total of 57 estimates (variances/covariances, direct, indirect and total effects; all reported in detail in Appendix A; Table A3, Table A4 and Table A5), only the ones related to the longitudinal PFE coping mediation are illustrated in Figure 3 and discussed further.

As with the GEE model, the standardized beta weights shown in Figure 3 suggest no direct association between baseline levels of screen time and depressive symptoms twelve months later (direct C-path: *beta* = −0.018; *p* = 0.280). However, a significant indirect effect (pathway) was apparent via PFE levels at the intermediate timepoint (indirect C’-path: *beta* = 0.001; *p* = 0.018). In other words, PFE was a significant and critical mediator between screen time levels at baseline and depressive symptoms after twelve months. Higher levels of screen time conferred a more detrimental depression trajectory only when it first had a detrimental effect on PFE levels during the first three months of the study. Note, however, that this did not hold true for the reversed causal direction, as Figure 3 shows that baseline depression did not predict screen time scores at the end of the study, neither directly (*beta* = 0.007; *p* = 0.767) nor indirectly (*beta* = −0.001; *p* = 0.771). In fact, no pathway shown in Figure 3 is statistically significant, except for the mediated C’-path (the B-path was almost significant: *p* = 0.063). The absence of a direct effect largely precludes an interpretation of the effect size in terms of a proportional mediation effect (calculated as the indirect effect divided by the total effect). Instead, the structural regression analysis was only able to isolate a longitudinal mediation effect, which by many standards can be said to be small in size (*std. beta* = 0.001; *p* = 0.018). Nonetheless, the effect was robust against non-normality (tested in 50,000 bootstraps), autocorrelation and time-related disturbances, as well as stochastic missing data imputations.

## 4. Discussion

### 4.1. Main Findings

A major knowledge gap in mental health-related screen time research regards causal mechanisms [7]. Therefore, longitudinal studies and examinations of moderating and mediating associations have become increasingly popular to investigate developments, pathways and trajectories of depression [21,23]. The main finding of our longitudinal analysis was that screen time alone was not associated with increased depressive symptoms at twelve-month follow-up when adjusted for gender and perceived socioeconomic status. However, in our primary analysis, we did find that problem-focused engagement coping (PFE) moderated the effect of screen time on depression. This was corroborated by a post hoc pathway analysis, which also suggested an indirect effect of screen time on subsequent depression, mediated by altered PFE coping levels in the three-month period between baseline and the first follow-up. However, consistent with previous research findings (e.g., [21]), depression did not appear to interfere with PFE in a way that affected future screen time; thus, no evidence of reversed causation was found. Together, these findings support the hypothesis of a causal or at least temporally sequential association between screen time, problem-focused engagement coping and depressive symptoms. Simply put, it appeared that individuals who engaged more in PFE coping were more resilient to the depression-related consequences of screen time, whereas individuals who engaged less in PFE coping were more likely to exhibit depressive symptoms because of increased screen time.

We also observed a well-known gender pattern in depression, where girls exhibited more symptoms than boys [1], and they also reported more EFE coping compared to boys [41]. However, we found no main effect nor screen time interaction effect involving emotion-focused engagement coping (EFE) at all. The gender-based findings related to coping are largely in line with previous research of coping in adult populations [41,44]. Our observation that boys had lower (i.e., “better”) PFE scores than girls both in the absolute and the relative sense, and not just relative to their EFE scores (Figure 2), could possibly be explained by the notion that personality-related differences in coping, especially in problem solving and cognitive restructuring, are often larger in adolescents than in adults [43].

### 4.2. Theoretical Explanations

One theory of PUI and harmful screen use is the “reinforcing spirals hypothesis”. It states that some screen activities, especially social media and other platforms that heavily rely on user recommendation algorithms, create “echo chambers” that tend to provide the user with information that reinforces already existing beliefs instead of challenging them [23,24,25]. For example, echo chambers have been hypothesized to amplify upward social comparisons, such as beauty ideals, and studies have indicated that social media use can affect people’s self-esteem and behaviors related to self-esteem, such as food choices [23,24]. With regards to the current study, echo chambers might account for the association between higher screen time and lower problem-focused engagement coping, possibly by filtering out cognitively dissonant information that could otherwise have promoted healthy cognitive restructuring and problem-solving behaviors. However, echo chambers do not necessarily hinder emotional expression or social contact, which are the main components of the socioemotional coping engagement style (EFE; as defined by [52,53]), because they offer a socially acceptable and often homogenous environment where relevant socioemotional needs are more easily fulfilled.

Another theoretical explanation to the health effects of screen time involves the “displacement hypothesis”, which posits that negative effects occur when screen-related behaviors substitute other healthy behaviors, for example, social contact or effortful problem solving. From a displacement perspective [21,22], one could speculate that screens do not merely offer immediate distraction from stressors (“escapism”) but, furthermore, decrease a person’s long-term need to practice certain problem-solving and emotionally regulatory skillsets that are only applicable in real life (IRL; e.g., ability to interpret subtle body language cues during social interaction, or the ability to sit through lengthy school lectures without looking at your phone several times).

Although this study cannot confirm that girls, compared to boys, were more prone to let screen time interfere with their PFE coping behaviors, the fact that they had lower PFE levels could potentially explain why some studies have found that girls are more sensitive towards screen-time-related mental health consequences [11,33]. Although girls generally had equal levels of EFE and PFE coping in our study (Figure 2), the former coping style (EFE) was not found to interact with screen time when controlling for gender and PFE interactions. This suggests that girls’ relatively high EFE coping levels did not constitute a compensatory protective effect, which is in line with previous studies showing that EFE coping is more frequently used by girls and women [41], and differently associated with subclinical depression in the normal population, depending on gender [44]. Since the present study did not investigate the role of different screen content (e.g., social media content) or disengagement coping styles (e.g., problem avoidance or self-criticism) we also cannot fully rule out the importance of EFE coping in the context of screen time. Rather, our results support the notion that screen time interferes with emotion regulation at the cognitive level, and to a lesser extent on a socioemotional level. However, several unmeasured biological and psychological factors may also account for the coping and depression patterns observed in this study [41,42,43].

In contrast to the idea that screens displace time, different screen and internet content seems to have different motivational “pull” (e.g., [63]). However, it is not fully understood how or why motivations to use screen devices lead to mental health problems [21]. It has been suggested that screen time motivations and desires can be viewed from a self-determination theory perspective, proposing that users seek out certain online and offline content because they fulfill basic psychological needs, related to autonomy, competence and relatedness [63,64]. This perspective may help explain why humans seek out social echo chambers online, i.e., because they can be perceived as psychologically “comforting”, and facilitate emotion regulation.

Escapism and other types of avoidant coping are particularly associated with unhealthy screen use and PUI. It is also generally accepted that screens have addictive properties [15], and individuals who spend large amounts of time with their screen devices tend to report stronger proneness to addiction, more internalizing symptoms and less effective use of coping strategies [27,64,65,66,67]. In turn, this might correspond to neurological brain activity in the amygdala, insula, striatum and parts of the prefrontal cortex, such as the dorsolateral prefrontal cortex, that affect decision making and attention towards or away from negative consequences of screen behaviors [15]. However, such behavioral patterns and brain functions are not only applicable to pathological use of the internet (PUI) and excessive use of internet-connected screen technology, but could equally well explain why this study observed similar effects in a normal adolescent population (although perhaps to a lesser extent).

### 4.3. Strengths and Limitations

One strength of this study is related to the longitudinal design as well as the large sample size of N = 4793 adolescents. Consistent with previous research (e.g., [21]), our longitudinal analyses of the association between screen time and depressive symptoms produced more conservative effects (*b* = 0.01) compared to our cross-sectional correlations (*r* = 0.21). However, the longitudinal analyses arguably produced less biased results as they account for baseline characteristics. Moreover, we used previously validated scales to measure depression and coping styles, although we were not able to differentiate between screen-based coping from other (offline) types of coping. At the same time, as with previous studies, our longitudinal design was associated with a certain degree of attrition and missing data, which despite sophisticated data imputation techniques, can confer some bias in estimates (e.g., bias related to self-selection processes during follow-up data collections).

Screen time was measured by an index of three items that primarily aimed to capture internet-based leisure screen time on both weekdays and weekends, excluding school-related tasks. Although the scale had quite acceptable internal consistency, it has not been previously validated and we cannot exclude the possibility that measurement errors somehow influenced the results of our analyses. This methodological issue is present in many, if not the majority, of screen time studies relying on self-reports, and the lack of standardized measurements points to an important but challenging goal for future research [7,16,28,68,69]. Moreover, our measurement aimed at quantifying the hours spent using any internet-connected device, without much consideration as to which screen type was used (e.g., computers, smartphones, tablets, smart TVs) or which screen content was viewed (e.g., gaming content vs. social media). Including such information in our statistical models could have changed our findings or brought more nuance to them. For example, a systematic review indicates that watching TV is less strongly associated with internalizing problems compared to other types of screen devices [35]. Screen device type and content preferences also generally differ by gender, where girls tend to engage more in smartphones and social media, while boys engage in computers and gaming [11]. We have previously shown that different types of screen content have varying mental health effects, but even so, our current choice of measurement was partially based on the finding that screen time still explained unique variance in depressive symptoms in models that adjusted for seven various online activities [27]. However, controlling for those activities did not fully account for gender differences in psychopathology, and would probably not do so in the current study either.

### 4.4. Implications for Practice and Prevention

Congruent with public health models [70], our results indicate that screen time may have stable (statistically significant) negative effects on mental health resilience in adolescent populations, even if the effect sizes are small. From the present results, it seems possible that some mental health problems related to consequences of screen time could be prevented by decreasing its likelihood of impacting problem-focused coping (i.e., problem solving and cognitive restructuring). Future research into different prevention strategies would, however, be needed to clarify if this could be achieved and what strategies are efficient. Moreover, the current collective evidence of screen time effects could be used to raise public awareness of how screens may be used in a healthier way, and to inform public health interventions aiming to promote healthy coping behaviors and prevent mental health problems (e.g., through universal prevention programs; [45]), rather than treating them within the health care system (e.g., through cognitive behavioral therapy; [71]). The evidence could also be used to develop policies regarding smartphone use and other types of screen use in schools and environments where the capability and readiness to engage in problem solving and cognitive restructuring is essential.

## 5. Conclusions

In the Swedish adolescent population, leisure screen time levels can interfere with problem-focused engagement coping and thereby elevate depressive symptoms over time. This study shows that it may be the case even in “healthy” adolescents if there is an impact on problem or emotional engagement coping. The effect size outside the pathological (PUI) context is probably small (maximum 3.4 BDI-II scores in this study), but mitigation of adverse influences on problem solving, cognitive restructuring ability and similar behaviors, could potentially improve the general mental health in youth populations, for example, through public health prevention programs.

## Figures and Tables

**Figure 1 ijerph-20-03771-f001:**
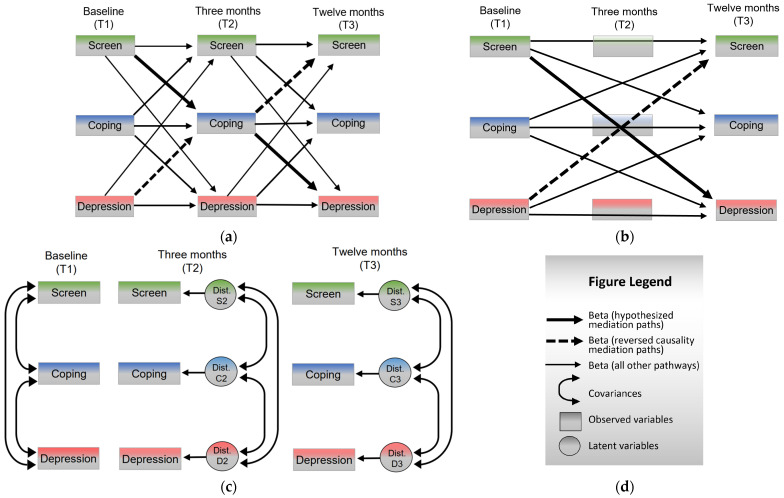
Conceptual illustration of the structural regression (pathway) model, divided into three parts to avoid clutter. Panel (**a**) shows indirect effects regression paths; panel (**b**) shows direct effects regression paths; panel (**c**) shows covariances and disturbance (error) terms; panel (**d**) shows the figure legend. The model was run unadjusted for gender and perceived socioeconomic status.

**Figure 2 ijerph-20-03771-f002:**
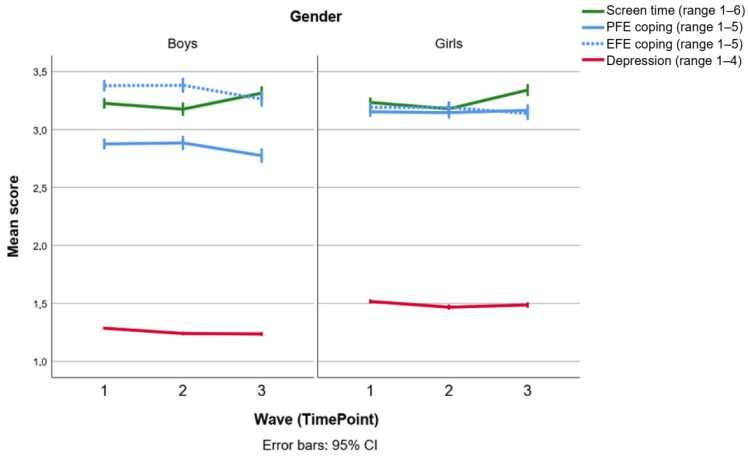
Observed mean scores and 95% confidence intervals for the four key variables, separately for each wave or timepoint (1 = baseline; 2 = three months; 3 = twelve months) and separately for boys (*n* = 2431) and girls (*n* = 2362). Note that problem-focused engagement (PFE) and emotion-focused engagement (EFE) coping scales are reversed such that higher scores indicate less coping (i.e., an alleged “risk factor”).

**Figure 3 ijerph-20-03771-f003:**
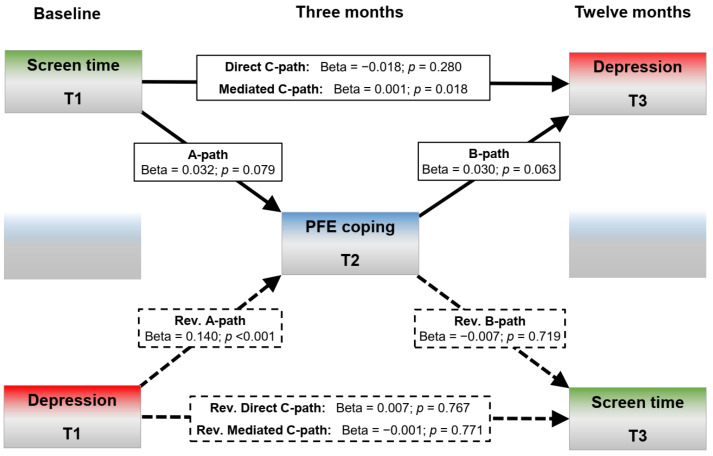
Structural regression (pathway) analysis and mediation test, with problem-focused engagement coping (PFE) as intermediate variable (50,000 bootstraps). Coefficients are standardized beta weights.

**Table 1 ijerph-20-03771-t001:** Results of Generalized Estimating Equation (GEE) model, and the primary hypothesis test of time × screen time × coping interactions. The PFE and EFE coping scales are reversed (lower scores indicate more coping). Far right column shows the coefficient effect size in terms of predicted depression score (PVMR) differences.

Parameter	*b*	95% CI (Wald)	S.E.	*p*	PVMR Difference ^1^
Intercept	0.347 *	0.295 to 0.400	0.0268	<0.001	
TIME = 3	−0.029 *	−0.047 to −0.01	0.0093	0.002	−0.055
TIME = 2	−0.012	−0.028 to 0.003	0.0079	0.125	n.s.
TIME = 1 (ref)	Ref.	Ref.	Ref.	Ref.	Ref.
(Gender = Girls)	0.131 *	0.119 to 0.142	0.0059	<0.001	+0.201
(Gender = Boys)	Ref.	Ref.	Ref.	Ref.	Ref.
Socioeconomic status (SES)	−0.063 *	−0.071 to −0.054	0.0041	<0.001	1-unit −0.100; Max-Min −0.399
Screen time	0.005	−0.009 to 0.019	0.007	0.469	n.s.
Problem-Focused Coping (PFE)	0.030 *	0.014 to 0.046	0.0081	<0.001	1-unit +0.097; Max-Min +0.292
Emotion-Focused Coping (EFE)	−0.001	−0.016 to 0.014	0.0074	0.895	n.s.
(Time = 3) × Screen time × PFE	0.009 *	0.003 to 0.015	0.0029	0.003	1-unit +0.065; Max-Min +0.447
(Time = 2) × Screen time × PFE	0.009 *	0.003 to 0.014	0.0026	0.001	1-unit +0.057; Max-Min +0.375
(Time = 1) × Screen time × PFE	0.011 *	0.006 to 0.017	0.0027	<0.001	1-unit +0.076; Max-Min +0.519
(Time = 3) × Screen time × EFE	−0.002	−0.007 to 0.004	0.0027	0.559	n.s.
(Time = 2) × Screen time × EFE	−0.003	−0.008 to 0.002	0.0024	0.263	n.s.
(Time = 1) × Screen time × EFE	−0.003	−0.008 to 0.002	0.0024	0.262	n.s.

* Statistically significant (*p* < 0.01) unstandardized beta coefficient, predicting raw depression scores (range 1–4) on logarithmic scale. ^1^ PVMR diff. = the difference in predicted raw depression scores (i.e., Predicted Value of Mean of Responses; PVMR) between levels of a variable. For continuous variables, both the average difference per 1.0-unit increment and the most extreme difference effect (Max-Min) is reported.

**Table 2 ijerph-20-03771-t002:** Heatmap illustrating the interaction between screen time and problem-focused engagement coping (PFE) across all three timepoints. Based on the GEE regression model, cell values show the pooled mean PVMR (Predicted Value of Mean of Response; i.e., raw depression scores ranging 1–4) for a certain range of values. The remaining variables (time, gender, SES, EFE) vary both within and between cells.

PVMR Values ^1^	Screen Time 1–1.99	Screen Time 2–2.99	Screen Time 3–3.99	Screen Time 4–4.99	Screen Time 5–6.00
PFE 1–1.99	1.18	1.19	1.20	1.22	1.25
PFE 2–2.99	1.23	1.26	1.29	1.32	1.35
PFE 3–3.99	1.30	1.34	1.39	1.43	1.49
PFE 4–5.00	1.40	1.42	1.48	1.55	1.63

^1^ Green color = small values; yellow color = medium values; red color = large values.

## Data Availability

Aggregated data are available in Appendix A, including correlation matrices and all individual parameters estimated in the structural regression. To protect the integrity of the research participants, the raw data are not publicly available. Methodological information to support replication by independent researchers is available from the corresponding author upon request.

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
