# Peer review of "Longitudinal Effects of Screen Time on Depressive Symptoms among Swedish Adolescents: The Moderating and Mediating Role of Coping Engagement Behavior"

_ijerph, 2023, doi:10.3390/ijerph20043771_

Round 1

Reviewer 1 Report

This is a well-written manuscript that explores relationship between electronic screen time and depression in a Swedish adolescent population. The authors clearly state the rationale for their study, describe appropriate methodology, and expound well-supported conclusions.

General concerns:

1.     Gender-based differences in the screen-time activities were noted.  Could the gender-based findings be explained solely by the screen-time activities?

2.     Gender-based differences in coping strategies were identified. Are these truly gender-based, or could it be that the screen-time activities influence the development of coping strategies? Does the literature give any hint as to which is more important?

Specific concerns:

1.     There are a few instances when the noun-verb tense is not appropriately synchronized (singular noun with a plural verb or vice versa)

2.     The acronym “PUI” is spelled “PIU” in line 74.

Reviewer 2 Report

This study explores the longitudinal effects of screen time on depressive symptoms among Swedish Adolescents and identifies a causal mechanism, which is interesting, but several issues need to be clarified.

First of all, what is the basis for this study to divide the coping styles to PFE and EFE? The different coping styles and the theoretical basis for taking PFE and EFE as mediating variables should be discussed and explained based on previous studies.

Secondly, reinforcing spirals hypothesis” and displacement hypothesis” is suggested to be introduced in the “Introduction”.

Finally, there are some mistakes in the manuscript, such as line 74, "PIU". It is recommended to proofread the entire manuscript.
